# Increased PARP Activity and DNA Damage in NSCLC Patients: The Influence of COPD

**DOI:** 10.3390/cancers12113333

**Published:** 2020-11-11

**Authors:** Jun Tang, Víctor Curull, Xuejie Wang, Coral Ampurdanés, Xavier Duran, Lara Pijuan, Alberto Rodríguez-Fuster, Rafael Aguiló, José Yélamos, Esther Barreiro

**Affiliations:** 1Pulmonology Department, Lung Cancer and Muscle Research Group, Hospital del Mar-IMIM, Parc de Salut Mar, Health and Experimental Sciences Department (CEXS), Universitat Pompeu Fabra (UPF), Medical School, Universitat Autònoma de Barcelona, Parc de Recerca Biomèdica de Barcelona (PRBB), 08003 Barcelona, Spain; jun.tang2@e-campus.uab.cat (J.T.); VCURULL@PARCDESALUTMAR.CAT (V.C.); Xuejie.Wang@e-campus.uab.cat (X.W.); 2Centro de Investigación en Red de Enfermedades Respiratorias (CIBERES), Instituto de Salud Carlos III (ISCIII), 08003 Barcelona, Spain; 3Cancer Research Program, Hospital del Mar Medical Research Institute (IMIM)-Hospital del Mar, 08003 Barcelona, Spain; campurdanes@imim.es (C.A.); jyelamos@imim.es (J.Y.); 4Scientific, Statistics and Technical Department, Hospital del Mar-IMIM, Parc de Salut Mar, 08003 Barcelona, Spain; xduran@imim.es; 5Pathology Department, Hospital del Mar-IMIM, Parc de Salut Mar, 08003 Barcelona, Spain; LPIJUAN@PARCDESALUTMAR.CAT; 6Thoracic Surgery Department, Hospital del Mar-IMIM, Parc de Salut Mar, 08003 Barcelona, Spain; ARodriguezFuster@parcdesalutmar.cat (A.R.-F.); RAGUILO@PARCDESALUTMAR.CAT (R.A.)

**Keywords:** lung cancer, DNA damage, PARP-1 and PARP-2 expression, PARP activity, COPD

## Abstract

**Simple Summary:**

Many people still die of lung cancer (LC), a disease that is mainly related to cigarette smoking. Smokers may also develop chronic obstructive pulmonary disease (COPD). COPD is a risk factor per se for LC. Cigarette smoking and other chemicals injure DNA on a daily basis. A repair mechanism based on PARP-1 and PARP-2 activity can restore damaged DNA to keep cells alive. However, cancer cells also take advantage of this mechanism to survive. Fifteen LC-COPD and 15 LC patients were enrolled in this study to elucidate whether COPD influences DNA damage-dependent PARP activity in lung tumors. DNA damage, PARP activity, PARP-1 and PARP-2 expression were analyzed in tumor and non-tumor lungs obtained during surgical resection of the lung tumors. DNA damage and PARP activity were increased only in tumors in LC-COPD patients. However, PARP-1 and PARP-2 expression decreased in tumors of both patient groups. LC patients with COPD may benefit from PARP inhibitor therapies.

**Abstract:**

(1) *Background*: Lung cancer (LC) is a major leading cause of death worldwide. Poly (ADP-ribose) polymerase (PARP)-1 and PARP-2 are key players in cancer. We aimed to assess PARP-1 and PARP-2 expression and activity and DNA damage in tumors and non-tumor lungs from patients with/without chronic obstructive pulmonary disease (COPD). (2) *Methods*: Lung tumor and non-tumor specimens were obtained through video-assisted thoracoscopic surgery (VATS) in LC patients with/without underlying COPD (two groups of patients, *n* = 15/group). PARP-1 and PARP-2 expression (ELISA), PARP activity (PARP colorimetric assay kit) and DNA damage (immunohistochemistry) levels were identified in all samples. (3) *Results*: Both PARP-1 and PARP-2 expression levels were significantly lower in lung tumors (irrespective of COPD)compared to non-tumor specimens, while DNA damage and PARP activity levels significantly increased in lung tumors compared to non-tumor specimens only in LC-COPD patients. PARP-2 expression was positively correlated with smoking burden in LC-COPD patients. (4) *Conclusions*: In lung tumors of COPD patients, an overactivation of PARP enzyme was observed. A decline in PARP-1 and PARP-2 protein expression was seen in lung tumors irrespective of COPD. Other phenotypic features (airway obstruction) beyond cancer may account for the increase in PARP activity seen in the tumors of patients with underlying COPD.

## 1. Introduction

Lung cancer (LC) is a leading cause of death worldwide [1,2,3,4]. Surgical resection of lung tumors continues to be the main elective treatment in LC patients [5,6,7]. However, this approach cannot be applied to patients with advanced stages. Other options such as chemotherapy and immune therapy with or without other biological or pharmacological agents are indicated in these cases [3]. 

Several mechanisms and risk factors are involved in the pathophysiology of non-small cell LC (NSCLC) [6]. Underlying respiratory conditions favor the incidence of lung tumors in patients, especially in those with emphysema [8,9]. Mechanisms such as inflammation, oxidative stress, epigenetics alterations, and the tumor microenvironment underlie the pathophysiology of tumor development in patients with chronic respiratory disorders including stromal structures and immune cell profiles, and have an impact on the patients’ survival [10,11,12]. Indeed, chronic obstructive pulmonary disease (COPD) is a major risk factor for LC development [9,13].

Interestingly, poly (ADP-ribose) polymerase-1 (PARP-1) has also been shown to play a significant role in elastase-induced lung inflammation and emphysema in a mouse model of COPD [14]. PARP-1 and PARP-2 catalytically cleave NAD^+^ and transfer ADP-ribose moieties onto specific amino residues of acceptor proteins in response to DNA damage. This process, termed poly ADP-ribosylation (PARylation), forms poly (ADP-ribose) polymers (PAR), which vary in size and branching. PAR elicit functional and structural changes in the target proteins [15,16,17]. PARP activity is upregulated in response to DNA damage to maintain DNA stability, integrity, and repair [18]. Accordingly, PARP inhibitors have emerged as promising therapeutic tools in cancer. They may act as potentiators of chemotherapeutic agents, immune therapy, and radiotherapy [19]. They can also be administered alone in tumors characterized by breast cancer (BRCA) gene mutations [20]. Accordingly, PARP inhibitors are currently in use for the treatment of several cancer types such as breast and ovarian cancer [21]. In small cell lung cancer (SCLC), the combination of PARP inhibitors and platinum based-chemotherapy showed superior efficacy compared to chemotherapy alone in a preclinical model [22]. Whether PARP-1 and PARP-2 expression and activity are also involved in LC development in patients with underlying COPD remains an open question.

On this basis, we hypothesized that PARP-1 and PARP-2 expression and activity may be increased in lung tumors of patients with COPD. Thus, we explored: (1) DNA damage, (2) PARP-1 and PARP-2 protein expression and PARP activity, and (3) correlations between clinical and biological variables in lung tumors of patients with and without COPD. A group of LC cancer patients with no COPD was included as a control group for the purpose of comparison. 

## 2. Results

### 2.1. Clinical and Functional Characteristics of Study Patients

The clinical and functional characteristics of all the study patients are shown in Table 1. Anthropometric variables such as age, gender and BMI did not show any significant difference between LC and LC-COPD patients. The number of male patients in the LC-COPD group was significantly larger than that in the LC group (Table 1). The cigarette smoking burden variable, pack-years was significantly higher in LC-COPD patients than in LC patients, while the number of non-smokers was significantly larger in the LC group (Table 1). The lung functional parameters FEV_1_, FEV_1_/FVC, DL_CO_ and K_CO_ were significantly reduced in LC-COPD patients compared to LC patients (Table 1). The majority of the COPD patients were in GOLD stages I and II (86.6%). TNM staging or histological subtypes did not significantly differ between the study groups. The levels of blood parameters such as total leucocytes, neutrophils, lymphocytes, albumin, total proteins and body weight loss did not significantly differ between the study patients.

### 2.2. DNA Damage Increased in Lung Tumors of COPD Patients

In LC-COPD patients, the percentage of γ-H2AX positive cells, a marker of DNA damage, was significantly higher in the tumors compared to non-tumor lung samples (Figure 1). On the contrary, the levels of DNA damage did not significantly differ between tumor and non-tumor specimens in LC patients (Figure 1).

### 2.3. PARP Activity Increased in Lung Tumors of COPD Patients

Consistent with the DNA damage results, a significant rise in PARP activity was also detected in lung tumor specimens of LC-COPD patients compared to control non-tumor samples (Figure 2). However, in LC patients, no significant differences in PARP activity were seen between tumor and non-tumor specimens (Figure 2).

### 2.4. PARP Expression Decreased in Lung Tumors of All the Study Patients

A significant decline was observed in PARP-1 and PARP-2 protein expression levels in tumors of both LC and LC-COPD patients compared to the respective non-tumor samples in both groups (Figure 3).

### 2.5. Influence of Staging in PARP Activity and Expression in LC and LC-COPD Patients

Interestingly, the variable staging of lung tumors did not significantly influence the PARP activity levels detected in the lung tumors when all the patients were analyzed as a whole (Figure 4).

The influence of LC staging in the PARP protein expression levels found in tumor specimens of all of the patients taken as a whole, revealed that patients in stage I exhibited the greatest amount of PARP-1 expression compared to the other LC stages (Figure 5A). Importantly, these results were also found in the LC patients with underlying COPD (Figure 5B), while PARP-1 expression was not modified according to staging in the group of LC patients (Figure 5C).

The LC staging variable did not influence lung tumor PARP-2 expression when all the patients were analyzed as a whole (Figure 6A). Additionally, no significant associations were found between LC staging and PARP-2 protein content in the lung tumors in either LC-COPD (Figure 6B) or LC patients (Figure 6C). 

### 2.6. PARP-2 Expression in Lung Tumors of COPD Patients Correlates with Cigarette Smoking

A significant positive correlation was found between the burden of cigarette smoking as indicated by the variable, pack-years and levels of PARP-2 expression in lung tumors of LC patients with underlying COPD (Figure 7).

## 3. Discussion

In tumors of patients with COPD compared to those of patients with no COPD, a rise in DNA damage and PARP activity was observed, while PARP-1 and PARP-2 protein expression levels decreased. The tumors of LC-COPD patients with stage I exhibited greater expression levels of PARP-1 enzyme whereas in patients with LC-only, no differences were seen in either PARP-1 or PARP-2 expression levels according to different stages. Moreover, PARP-2 expression in lung tumor specimens significantly correlated with cigarette smoking burden among LC-COPD patients.

PARP-1 is the most relevant enzyme of the PARP family as its activity accounts for 85–90% of poly (ADP-ribosyl)ation in cells [23,24]. PARP activity plays a significant role in DNA repair through the induction of post-translational modifications of the target proteins by the transfer of ADP ribose moieties using NAD^+^ as a substrate [23]. Other effects of the overaction of PARP include the occurrence of necrosis in tissues in response to persistent DNA damage. As such necrosis may take place as a result of depletion of the substrate NAD^+^ [25]. 

In the current investigation, levels of DNA damage, as measured by γ-H2AX significantly increased in the tumors of LC-COPD, suggesting that severe injury of the DNA took place in the tumor cells of these patients. Likewise, the overactivation of PARP activity took place only in the tumors of the same patients. Moreover, a significant positive correlation between pack-years and PARP-2 expression was only observed in the LC-COPD patients. These are relevant findings that suggest that chronic cigarette smoking induces DNA damage, which may be counterbalanced by PARP activity. In fact, DNA damage and PARP-1 overactivation induced the parthanatos pathway of cell death as a result of the exposure of human bronchial epithelial cells to cigarette smoke [26]. Moreover, in patients with stable COPD, systemic PARP-1 activation was also observed in the lymphocytes along with increased inflammation and oxidative stress [27].

Importantly, the expression of PARP-1 and PARP-2 was significantly lower (the decrease ranged from 44% to 52%) but did not disappear in the tumors of both groups of patients, irrespective of COPD. Interestingly, despite the reduced expression of PARP-1 and PARP-2, overall PARP activity was maintained and even significantly increased in the tumors of the patients with the underlying respiratory condition. These findings are in agreement with previous results, in which the protein content of PARP isoforms did not influence PARP activity in several cancer cell lines [28]. Biological mechanisms such as endogenous activation or repression and/or post-translational modifications may account for the lack of correlations between PARP enzyme activity and protein expression levels. 

Furthermore, the findings of the present investigation suggest that stimuli beyond the cancer phenotype were most likely part of the pathophysiology of the overaction of PARP activity. In keeping with this, elastase-induced emphysema was shown to increase PARP activity in mouse lungs in an experimental model of COPD [14]. Taken together, these results point to a potential role of PARP inhibitors in the treatment of LC patients, particularly in those with underlying respiratory conditions as tumors of those without COPD did not experience an increase in PARP activity. This scenario might account for the lack of significant beneficial effects of the PARP inhibitor, olaparib in combination with gefitinib in a phase 2 trial in NSCLC patients [29]. 

In summary, we studied two different groups of patients with LC. The differential phenotypic features of the patients associated with the underlying respiratory disease evidenced that these aspects should be taken into account when designing the best therapeutic strategies for the management of LC patients, especially as the burden of DNA damage and the activity of PARP enzymes were only significantly greater in the tumor specimens of patients with underlying COPD.

### Study Limitations

A potential limitation of this study was the relatively low number of patients analyzed in the investigation. Nonetheless, it should be mentioned that very selective inclusion and exclusion criteria were established to recruit the patients. Moreover, patients had to undergo video-assisted thoracoscopic surgery (VATS) for the resection of their lung neoplasm, which is not the case for all LC patients seen in specialized clinics. Additionally, sample size was calculated ad hoc by the statistician in the group. As such, a minimum of 13 patients/group was estimated to be necessary to fulfill the study objectives. Furthermore, as 15 patients/group were finally recruited, the power of the study was 83.90%, and PARP activity was the target variable. All the results obtained in the present investigation, including the association analyses are based on the analysis of the 15 patients recruited in each group. 

## 4. Materials and Methods

### 4.1. Study Design and Ethics

This is a cross-sectional, prospective study designed according to the World Medical Association guidelines (Seventh revision of the Declaration of Helsinki, Fortaleza, Brazil, 2013) for human subjects involved in medical investigations. The study was approved by the institutional Ethics Committee on Human Investigation (protocol # 2008/3390/I, at Hospital del Mar–IMIM, Barcelona, Spain). All the participants of the study signed the informed written consent.

Patients were prospectively recruited from the Lung Cancer Clinic at Hospital del Mar (Barcelona, Spain). All the participants were part of the Lung Cancer Mar Cohort. For the purpose of the current study, 30 patients with LC were recruited in 2019. Candidate patients for tumor resection underwent VATS prior to the administration of any kind of adjuvant therapy. Tumor and non-tumor lung specimens were collected from all the study participants.

LC diagnosis and staging were established by histological confirmation and classified in accordance with currently available guidelines for the diagnosis and management of LC [30,31]. TNM (tumor, node, and metastasis) staging was defined as stated in the 8th edition of the Lung Cancer Stage Classification [32]. COPD diagnosis was established as a post-bronchodilator forced expiratory volume in one second (FEV_1_)/forced vital capacity (FVC) ≤ 0.7, which is not fully reversible by spirometry according to currently available guidelines for diagnosis and management of COPD [33,34]. Exclusion criteria were: small cell lung cancer (SCLC), chronic cardiovascular disease, restrictive lung disease, metabolic, immune disease, or clot system disorders, signs of severe inflammation and/or bronchial infection (bronchoscopy), current or recent invasive mechanical ventilation, or long-term oxygen therapy. The presence/absence of these diseases was confirmed using standard clinical tests: clinical examination, blood tests, bronchoscopy, electrocardiogram, echocardiography, and exercise capacity evaluation. Patients were further subdivided post hoc into two groups based on the presence of COPD: (1) 15 LC patients with COPD (LC-COPD group) and (2) 15 LC patients without COPD (LC group). 

### 4.2. Clinical Assessment

Lung function parameters were evaluated according to standard procedures in all the study patients. In patients with underlying COPD, the diagnosis and severity were determined in accordance with currently available guidelines [35,36]. A nutritional evaluation was done for all patients including body mass index (BMI) and blood nutritional parameters. 

### 4.3. Collection and Preservation of Samples

Lung specimens were obtained from tumors and the surrounding non-tumor parenchyma following standard technical procedures during VATS for the standard care in the treatment of lung tumors. In all of the study patients, the expert pulmonary pathologist selected an approximately 10 × 10 mm^2^ area of tumor and non-tumor specimen from the fresh lung samples. Non-tumor specimens were obtained as far distal to the tumor margins as possible (average >7 cm). Fragments of both tumor and non-tumor samples were fixed in formalin and embedded in paraffin blocks until further use. Another fragment was frozen immediately in liquid nitrogen and preserved at −80 °C for the measurement of protein levels.

### 4.4. Molecular Biology Analysis

#### 4.4.1. DNA Damage in Lungs of the Study Patients Using Immunohistochemistry

DNA damage was assessed in tumor and non-tumor lung specimens by the presence of γ-H2AX, a hallmark of DNA damage [37] using conventional immunohistochemistry as previously described [10,38]. Briefly, paraffin-embedded specimens were cut into three-micrometer sections on a microtome. Following deparaffinization, lung sections were submerged in pre-heated antigen-retrieval solution of in Citrate Buffer (PH = 6) in a pressure cooker for 15 min and then slides were gradually cooled to room temperature. After rinsing with distilled water three times, slides were treated with 3% hydrogen peroxide for ten minutes to block endogenous peroxidase activity. Then, slides were incubated with blocking buffer (PBS 1% bovine serum albumin) for one hour at room temperature and with anti-γ-H2AX primary antibody (anti-γ-H2AX, Millipore) at 4 °C overnight. The next day, after being washed three times with PBS, the slides were incubated with biotinylated universal secondary antibody for one hour and the detection process was assessed using HRP-streptavidin for five minutes. After two minutes of hematoxylin counterstaining, slides were dehydrated and mounted with dibutylphthalate polystyrene xylene (DPX) mounting medium for conventional microscopy examination. Images of the stained tumor and non-tumor lung sections of patients were obtained under a light microscope (Olympus, Series BX50F3, Olympus Optical Co., Hamburg, Germany) coupled with an image-digitizing camera (Pixera Studio, version 1.0.4, Pixera Corporation, Los Gatos, CA, USA). In addition, anti-γ-H2AX positively stained cells in all of the samples were counted independently by two previously trained investigators. The area of the lung sections was measured using Image J software (National Institutes of Health, Bethesda, MD, USA) in all stained slides. Data are presented as the percentage of anti-γ-H2AX cells in the measured area in all the non-tumor and tumor lung specimens of patients.

#### 4.4.2. PARP Activity in Lungs of the Study Patients

PARP enzyme activities in human lung samples were estimated using the higher throughput (HT) 96 test size Universal Colorimetric PARP Assay Kit with Histone-Coated Strip Wells (Trevigen, Gaitherburg, MD, USA) according to the manufacturer’s instructions and previous studies [39]. The amount of total protein levels in lung homogenates were quantified using NanoDrop ND1000 spectrophotometer (Thermo Scientific, Waltham, MA, USA) with duplicates of each sample. Briefly, all histone-coated strip wells were rehydrated with PARP buffer for 30 min. A standard curve was always run with each assay along with the samples. The identical amounts of proteins (80 µg) from lung homogenates were added and incubated with 25 µL PARP cocktail (mixture of PARP cocktail, activated DNA and PARP buffer) for one hour at room temperature. Then, wells were washed twice with 1 × PBS and 0.1% Triton X-100 and twice with 1 × PBS and incubated with 50 µL Strep-HRP for one hour at room temperature. After four additional washes, wells were incubated in the dark with 50 µL TACS-Sapphire ^TM^ colorimetric substrates for fifteen minutes at room temperature. Lastly, the enzyme reactions were stopped by adding 50 µL of 0.2mol of hydrochloric acid per well. Absorbance was read in a microplate reader at 450 nm. Intra-assay coefficients of variation for PARP activity level were less than 4% and inter-assay coefficients of variation were less than 5%.

#### 4.4.3. PARP Expression in Human Lungs Using Enzyme Linked Immunosorbent Assay

PARP-1 and PARP-2 expression levels in tumor samples and non-tumor samples of human lungs were quantified using ELISA (MyBioSource, Inc., San Diego, CA, USA). All procedures were performed following the manufacturer’s instructions and previous studies [10]. The amount of total protein levels in lung homogenates were quantified using NanoDrop ND1000 spectrophotometer (Thermo Scientific, Waltham, MA, USA) with duplicates of each sample. Before starting the assay, reagents and samples were naturally warmed to room temperature. A standard curve was always run with each assay along with the samples. An identical volume of samples (40 µL) and biotinylated human PARP-1 and PARP-2 antibodies (10 µL) were added to the pre-coated wells with the correspondent PARP-1 and PARP-2 antibodies in duplicate. Subsequently, 50 µL streptavidin-HRP secondary antibodies were loaded into all wells and plates were incubated on an orbital micro-plate at 37 °C for one hour. Then, all wells were washed five times for 45 s and incubated for another ten minutes with substrate solutions at 37 °C in the dark. Finally, 50 µL stop solution was added to each well to end the enzyme reaction. Optical densities in each well were immediately detected in a microplate reader set to 450 nm. Intra-assay coefficients of variation were less than 8% and inter-assay coefficients of variation were less than 10% for both PARP-1 and PARP-2 levels.

### 4.5. Statistical Analysis

The normality of the study variables was tested using the Shapiro-Wilk test. Accepting an alpha risk of 0.05 and a beta risk of 0.2 in a two-sided test, 13 subjects were necessary in each group to recognize a minimum difference of 0.015 units in the mean of the variable PARP activity as statistically significant. The common deviation was assumed to be 0.011. Finally, as 15 patients were recruited in each group, by taking the results of the one-way analysis of variance for the variable PARP activity group means, that is, 0.0149, 0.0210, 0.0136, 0.0248, a variance error = 9.75 ×10^−5^ and a sample size = 15 (balanced groups) we obtained a power of 83.90%. 

Qualitative variables are represented as total numbers and percentages with respect to total values, while quantitative variables are reported as the mean and standard deviations. Differences in physiological variables in clinical parameters between LC and LC-COPD groups of patients were analyzed using Student’s T-test. Differences between patient groups (LC and LC-COPD) and types of samples (tumor and non-tumor) were analyzed using Kruskal–Wallis equality-of-populations rank test followed by Dunn’s Pairwise Comparison or one-way ANOVA and Tukey post hoc to adjust for multiple comparisons of the biological variables. Statistical significance was established at *p* ≤ 0.05. Physiological and clinical variables are shown in tables, while biological variables are in figures that use violin plots or histograms according to the normality of the variables. All the statistical analyses of the study were conducted using the software Statistical Package for the Social Science (SPSS, version 23, SPSS Inc., Chicago, IL, USA).

## 5. Conclusions

In lung tumors of patients with underlying COPD, an overactivation of PARP enzyme was observed along with increased DNA damage levels. A decline in PARP-1 and PARP-2 protein expression was seen in lung tumors irrespective of COPD. Other phenotypic features (airway obstruction) beyond cancer may account for the increase in DNA damage and PARP overactivation seen in the tumors of patients with underlying COPD. These findings warrant special attention when designing specific therapeutic strategies that may include PARP inhibitors in the treatment of patients with NSCLC as COPD may render these patients more prone to benefit from those therapies.

## Figures and Tables

**Figure 1 cancers-12-03333-f001:**
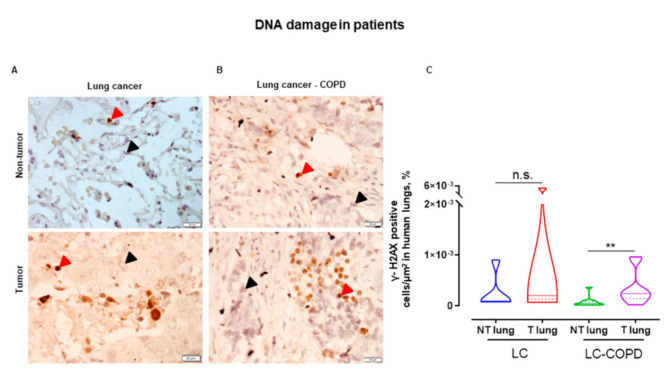
(**A**,**B**) Representative immunohistochemical staining sections of γ-H2AX in non-tumor and tumor lung specimens of LC and LC-COPD patients. Red arrows point towards γ -H2AX cells, which were positively stained in brown, and black arrows point towards γ -H2AX cells that remained stained in blue (hematoxylin counterstaining). (**C**) Violin plots with median (continuous line) and interquartile ranges (discontinuous lines) of the number of γ -H2AX positively stained cells in the total measured area. Statistical significance: n.s., no significance, ** *p* < 0.01 between non-tumor and tumor samples in LC-COPD patients. Definition of abbreviations: PARP, poly (ADP-ribose) polymerase; LC, lung cancer; COPD, chronic obstructive pulmonary disease.

**Figure 2 cancers-12-03333-f002:**
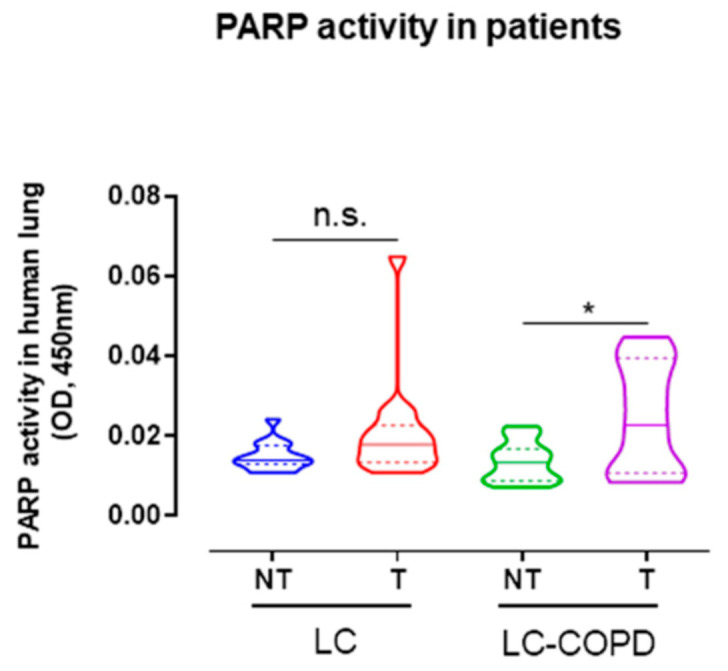
Violin plots with median (continuous lines) and interquartile ranges (discontinuous lines) of PARP activity levels in non-tumor and tumor lung specimens of LC and LC-COPD patients. Statistical significance: n.s., no significance, * *p* < 0.05 between non-tumor and tumor specimens of LC-COPD patients. Definition of abbreviations: PARP, poly (ADP-ribose) polymerase; OD, optical densities; LC, lung cancer; COPD, chronic obstructive pulmonary disease.

**Figure 3 cancers-12-03333-f003:**
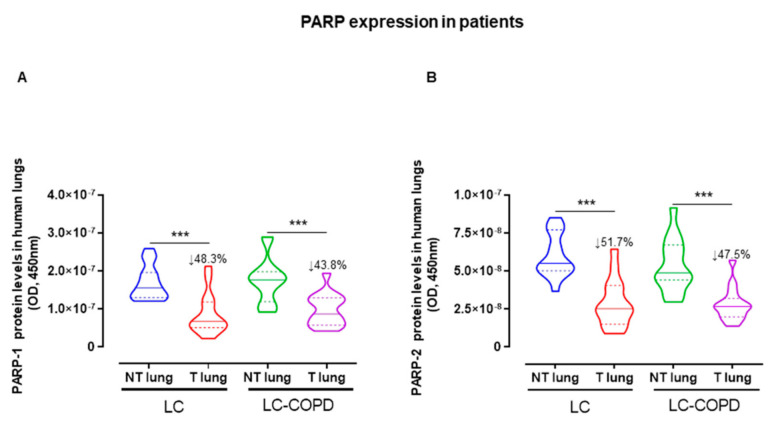
(**A**,**B**) Violin plots with median (continuous line) and interquartile ranges (discontinuous lines) of PARP-1 and PARP-2 protein levels assessed by ELISA in LC and LC-COPD patients, respectively. Statistical significance: *** *p* ≤ 0.001 between tumor (T) and non-tumor (NT) specimens in LC and LC-COPD groups. Definition of abbreviations: PARP, poly (ADP-ribose) polymerase; OD, optical densities; LC, lung cancer; COPD, chronic obstructive pulmonary disease.

**Figure 4 cancers-12-03333-f004:**
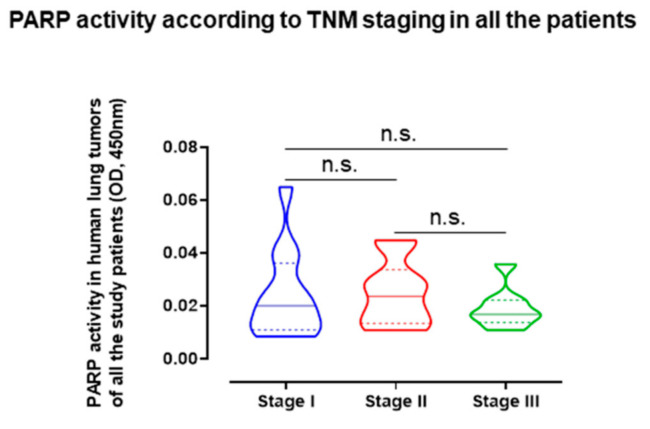
Violin plots with median (continuous lines) and interquartile ranges (discontinuous lines) of PARP activity levels in lung tumors according to cancer stages of all the study patients. Statistical significance: n.s., no significance among cancer stages of patients. Definition of abbreviations: PARP, poly-ADP ribose polymerase; OD, optical densities.

**Figure 5 cancers-12-03333-f005:**
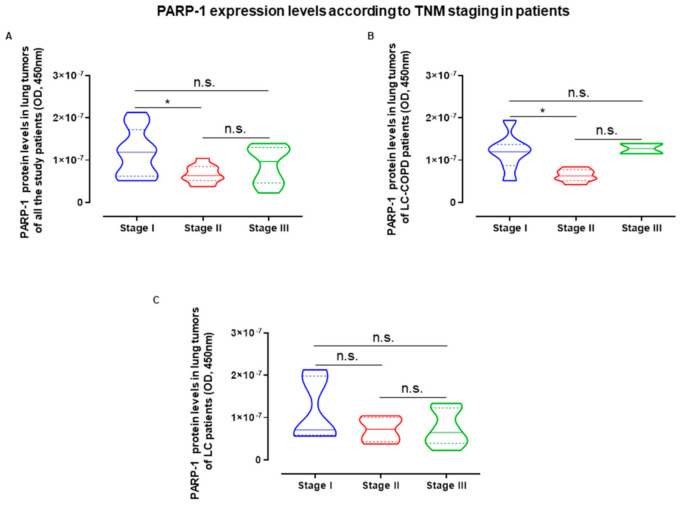
(**A**–**C**) Violin plots with median (continuous lines) and interquartile ranges (discontinuous lines) of PARP-1 protein levels in lung tumors according to cancer stages in all the study patients, LC-COPD and LC patients respectively. Statistical significance: * *p* < 0.05, n.s., no significance among cancer stages of patients. Definition of abbreviations: PARP, poly-ADP ribose polymerase; OD, optical densities.

**Figure 6 cancers-12-03333-f006:**
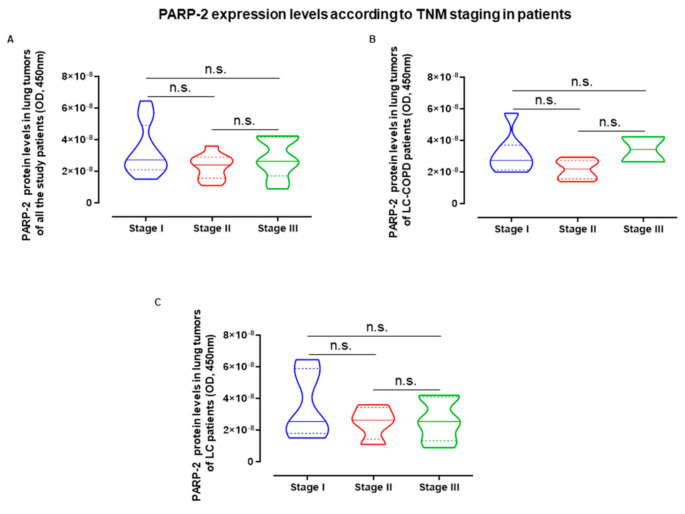
(**A**–**C**) Violin plots with median (continuous lines) and interquartile ranges (discontinuous lines) of PARP-2 expression levels in lung tumors according to cancer stages of all the study patients, LC-COPD and LC patients respectively. Statistical significance: n.s., no significance among cancer stages of patients. Definition of abbreviations: PARP, poly-ADP ribose polymerase; OD, optical densities.

**Figure 7 cancers-12-03333-f007:**
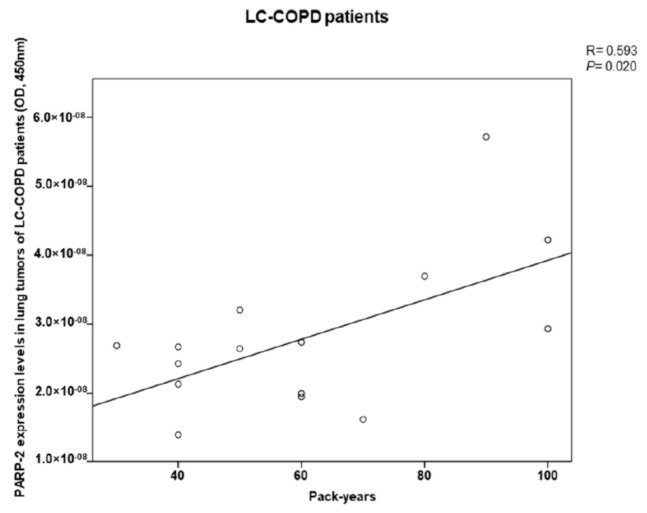
A significant positive correlation was detected between PARP-2 expression levels in lung tumors and pack-years in LC-COPD patients. Definition of abbreviations: poly (ADP-ribose) polymerase; LC, lung cancer; COPD, chronic obstructive respiratory disease.

**Table 1 cancers-12-03333-t001:** Clinical and functional characteristics of the study patients.

Anthropometric Variables	LC (*N* = 15)	LC-COPD (*N* = 15)
Age, years	64 (11)	67 (9)
Male, *N*/Female, *N*	6/9	10/5
BMI, kg/m^2^	27 (4)	28 (7)
Smoking history		
Current: *N*, %	5, 33.3	6, 40
Ex-smoker: *N*, %	5, 33.3	9, 60
Never smoker: *N*, %	5, 33.3	0 *
Pack-years	23 (23)	61 (23) ***
Lung function parameters		
FEV_1_	88 (23)	70 (20) *
FEV_1_/FVC, %	76 (5)	58 (11) ***
DLCO, %	85 (17)	57 (13) ***
KCO, %	83 (13)	55 (10) ***
GOLD stage		
GOLD stage I: *N*, %	NA	5, 33.3
GOLD stage II: *N*, %	NA	8, 53.3
GOLD stage III: *N*, %	NA	2, 13.3
GOLD stage IV: *N*, %	NA	0, 0
TNM staging		
Stage I: *N*, %	5, 33.3	7, 46.7
Stage II: *N*, %	4, 26.7	6, 40
Stage III: *N*, %	6, 40	2, 13.3
Histological diagnosis		
Squamous cell carcinoma: *N*, %	0, 0	0, 0
Adenocarcinoma: *N*, %	15, 100	15, 100
Others: *N*, %	0, 0	0, 0
Blood parameters		
Total leucocytes/μL	9.3 (3.5) × 10^3^	10.1 (4.4) × 10^3^
Total neutrophils/μL	6.8 (3.9) × 10^3^	7.5 (4.7) × 10^3^
Total lymphocytes/μL	1.8 (0.8) × 10^3^	1.8 (0.9) × 10^3^
Albumin (g/dL)	4.3 (0.4)	4.4 (0.5)
Total proteins (g/dL)	6.9 (0.4)	7.1 (0.6)
Body weight loss, kg		
0, *N*, %	13, 86.6	12, 80
1–5, *N*, %	1,6.7	1, 6.7
6–10, *N*, %	1,6,7	2, 13.3

Continuous variables are presented as mean and standard deviation while categorical variables are presented as the number of patients in each group and the percentage in the study group with respect to the total population. Definition of abbreviations: N, number; kg, kilograms; m, meters; BMI, body mass index; FEV_1_, forced expiratory volume in one second; FVC, forced vital capacity; DL_CO_, carbon monoxide transfer; K_CO_, Krogh transfer factor; GOLD: Global initiative for chronic Obstructive Lung Disease; NA, not applicable; TNM, tumor, nodes, metastasis; L, liter. Statistical analyses and significance: * *p* < 0.05, *** *p* < 0.001 between LC-COPD patients and LC patients.

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
