# Peer review of "Increased PARP Activity and DNA Damage in NSCLC Patients: The Influence of COPD"

_cancers, 2020, doi:10.3390/cancers12113333_

Round 1

Reviewer 1 Report

Main comments:

Since Figure 2 is described in the text before Figure 1, I suggest to the authors to correct the number of figures, transforming Figure 2 into Figure 1 and vice versa.

The results section needs to be described in more detail.

The Figure 3 must have the same format as the other figures. Please convert it to violin plots

In figure 5 the authors show the difference in terms of PARP1 expression between all the study patients, LC-COPD and LC patients. And PARP2? Please show in figure 6 the differences of PARP2 also between LC-COPD and LC patients.

The authors state: “In tumors of patients with COPD compared to those of patients with no COPD, a rise in DNA damage and PARP activity was observed, while PARP-1 and PARP-2 protein expression levels decreased”. In their opinion, how the reduction of protein expression is explained?

At page 10, line 268 the authors talk about lung of mice, although in the study there are no references to studies on mice. Please clarify this.

Minor comments

The frame style of the graphs should be represented as offset X and Y axes, in such a way as to make the violins better visible.

Page 12, line 324-325: incorrect format

Page 7, line 157-158: incorrect format

Author Response

REVIEWER #1:

C1

Comments and Suggestions for Authors

Main comments:

Since Figure 2 is described in the text before Figure 1, I suggest to the authors to correct the number of figures, transforming Figure 2 into Figure 1 and vice versa.

R1

We thank the reviewer for this remark. However, this was not the authors’ fault as the order of the description of the results was correct in the submitted file. We have modified the order of the text in the revised manuscript version (Journal’s template, see pages 3-5). We hope that the editorial office will keep the order of the results in place as submitted by the authors.  

We thank the reviewer for all the insightful comments. We have modified the manuscript text following the precise recommendations given by the two reviewers. Moreover, appropriate responses to each of the reviewer’s concerns are also given below.

C2

The results section needs to be described in more detail.

R2

We thank for this comment. We have improved the Results section in the revised manuscript (See modified text in lines 85-192, pages 3-10).

C3

The Figure 3 must have the same format as the other figures. Please convert it to violin plots

R3

We thank the reviewer for this comment. Figure 3 has been modified accordingly, see line 144 of page 6 in the revised manuscript.

C4

In figure 5 the authors show the difference in terms of PARP1 expression between all the study patients, LC-COPD and LC patients. And PARP2? Please show in figure 6 the differences of PARP2 also between LC-COPD and LC patients.

R4

We thank the reviewer for this comment. Accordingly, we have modified Figure 6 in the revised version, in which the results of each study group are depicted separately. See line 178 of page 9 in the revised manuscript.

C5

The authors state: “In tumors of patients with COPD compared to those of patients with no COPD, a rise in DNA damage and PARP activity was observed, while PARP-1 and PARP-2 protein expression levels decreased”. In their opinion, how the reduction of protein expression is explained?

R5

We thank the reviewer for this comment. Our findings are in line with those previously demonstrated by Zaremba et al (British Journal of Cancer 2009, reference # 28). The results obtained herein have been discussed in the revised Discussion (See lines 220-224 in the revised manuscript).

C6

At page 10, line 268 the authors talk about lung of mice, although in the study there are no references to studies on mice. Please clarify this.

R6

We thank the reviewer for this remark. No mice were studied in this investigation. We apologize for this mistake. The word has been deleted in the revised manuscript (see line 307 of the revised manuscript).

Minor comments

C7

The frame style of the graphs should be represented as offset X and Y axes, in such a way as to make the violins better visible.

R7

We thank the reviewer for this comment. Accordingly, all the manuscript figures have been modified in the revised manuscript.

C8

Page 12, line 324-325: incorrect format

R8

We thank the reviewer for this remark. However, this was not the authors’ mistake as we submitted a plain word document in which the font type and size was identical throughout the entire text. We have corrected the font and size type in the revised manuscript version (See lines 363-364 in the revised manuscript).

 C9

Page 7, line 157-158: incorrect format

R9

We thank the reviewer for this remark. However, this was not the authors’ mistake as we submitted a plain word document in which the font type and size was identical throughout the entire text. We have corrected the font and size type in the revised manuscript version (See lines 174-175 in the revised manuscript).

Reviewer 2 Report

The Authors evaluated PARP-1 and PARP-2 expression and activity and DNA damage in tumors and non-tumor lungs from 30 patients with/without chronic obstructive pulmonary disease (COPD). They found that an overactivation of PARP enzyme was observed in lung tumors of COPD patients, while a decline in PARP-1 and PARP-2 protein expression was seen in lung tumors irrespective of COPD. The findings are interesting and hypothesis generating.

I suggest the following revisions:

1) Materials and methods should be reported after introduction;

2) Figure 2 is cited in the text before figure 1...

3) The findings reported on the correlation of stage I and PARP expression (Figure 5 and 6) refer to few patients and should be deleted

4) The Authors should report in the discussion the limitations of the study

Author Response

REVIEWER #2:

C1

The Authors evaluated PARP-1 and PARP-2 expression and activity and DNA damage in tumors and non-tumor lungs from 30 patients with/without chronic obstructive pulmonary disease (COPD). They found that an overactivation of PARP enzyme was observed in lung tumors of COPD patients, while a decline in PARP-1 and PARP-2 protein expression was seen in lung tumors irrespective of COPD. The findings are interesting and hypothesis generating.

R1

We thank the reviewer for this comment and for having considered that our study is interesting and hypothesis generating. We have modified the manuscript text following the precise recommendations given by the two reviewers. Moreover, appropriate responses to each of the reviewer’s concerns are also given below.

C2

I suggest the following revisions:

1) Materials and methods should be reported after introduction;

R2

We thank the reviewer for this comment. Nonetheless, the Journal requires that this format is used instead of the classical format of Methods following Introduction. We cannot modify the order of these sections in the revised manuscript.

C3

2) Figure 2 is cited in the text before figure 1...

R3

We thank the reviewer for this remark. However, this was not the authors’ fault as the order of the description of the results was correct in the submitted file. We have modified the order of the text in the revised manuscript version (Journal’s template, pages 3-5). We hope that the editorial office will keep the order of the results in place as submitted by the authors. 

C4

3) The findings reported on the correlation of stage I and PARP expression (Figure 5 and 6) refer to few patients and should be deleted

R4

We thank the reviewer for this comment. Despite that the number of patients studied in this sort of analyses is relatively low, we still believe that they are relevant. Moreover, reviewer # 1 has requested that results corresponding to the analyses of the two study groups of patients separately are also included in Figure 6. Thus, Figures 5 and 6 have been kept in the revised manuscript version. The format of all the figures has been modified according to comments raised by reviewer # 1 (See R7 responses to reviewer # 1 above). Additionally, in the “Study limitations” section within the revised Discussion, the relatively reduced number of patients analyzed in Figures 5 and 6 has been mentioned (See lines 239-248 in the revised manuscript).

C5

4) The Authors should report in the discussion the limitations of the study.

R5

We thank the reviewer for this comment. Accordingly, the limitations of the study are described in lines 239-248 of the revised Discussion.